# A phenotypic screening platform utilising human spermatozoa identifies compounds with contraceptive activity

Franz S Gruber[1], Zoe C Johnston[1,2], Christopher LR Barratt[2]*, Paul D Andrews[1]

[1]National Phenotypic Screening Centre, School of Life Sciences, University of Dundee, Dundee, United Kingdom; [2]Reproductive and Developmental Biology, Division of Systems Medicine, School of Medicine, Ninewells Hospital and Medical School, University of Dundee, Dundee, United Kingdom

**Abstract** There is an urgent need to develop new methods for male contraception, however a major barrier to drug discovery has been the lack of validated targets and the absence of an effective high-throughput phenotypic screening system. To address this deficit, we developed a fully-automated robotic screening platform that provided quantitative evaluation of compound activity against two key attributes of human sperm function: motility and acrosome reaction. In order to accelerate contraceptive development, we screened the comprehensive collection of 12,000 molecules that make up the ReFRAME repurposing library, comprising nearly all the small molecules that have been approved or have undergone clinical development, or have significant preclinical profiling. We identified several compounds that potently inhibit motility representing either novel drug candidates or routes to target identification. This platform will now allow for major drug discovery programmes that address the critical gap in the contraceptive portfolio as well as uncover novel human sperm biology.

## Introduction

An effective and comprehensive family planning strategy is fundamental to delivery on the United Nations 17 Sustainable Development Goals (*Starbird et al., 2016*; *Goodkind et al., 2018*), however the current contraceptive portfolio is suboptimal. For example, it is estimated that more than 214 million women in developing countries have an unmet need for contraception which results in 89 million unintended pregnancies and 48 million abortions every year, with substantial impacts on maternal and child health and poverty levels (*Guttmacher Institute, 2017*). To achieve the UN targets it is critical that involvement of men be considered of equal importance as women (*Hardee et al., 2016*). However, it is noticeable that no effective, reversible and widely available form of contraception has been developed for men since the condom and as such the burden falls largely on the woman. The development of drugs that can be used in the male would address a critical gap in the contraceptive portfolio.

Progress in developing new male contraceptives has been slow for a plethora of reasons. Despite decades of research, safe and effective hormonal approaches to prevent spermatozoa production have yet to be realised and viable alternatives remain elusive (*Page and Amory, 2018*). Developing a drug for any purpose is difficult but one that blocks sperm function is a particular challenge as spermatozoa don't divide, transcribe or translate, have specialized organelles, are highly motile and continually produced in large numbers (~100 million per day). Moreover, as the production of fertilization competent cells is necessarily different between species, it is essential to use a human system early in the drug discovery cascade to increase the chances of translational success. The spermatozoa's sole function post-ejaculation is to swim until it closes in on the oocyte. The spermatozoa are

*For correspondence:
c.barratt@dundee.ac.uk

then able to undergo a unique exocytotic process called the acrosome reaction (AR) which facilitates penetration of the zona pellucida, the oocyte's outer glycoprotein shell and allows fusion with the oocyte's plasma membrane, delivering its DNA. Hence, a male contraceptive could work in a number of different ways – it could completely halt spermatogenesis, block an essential function of the spermatozoon (such a motility) or perturb a cellular process that results in a non-productive interaction with the oocyte (such as prematurely triggering the acrosome reaction, and/or by blocking exocytosis or inhibiting receptor binding). Low sperm motility is a common cause of male infertility (*Samplaski et al., 2019*) indicating that targeting this process should yield effective drugs. The options for delivery of a contraceptive are wide ranging, but our primary aim is to develop a relatively fast-acting safe drug, self-administered by the male, that specifically and effectively irreversibly blocks the sperm's ability to get to the egg and/or fertilise it.

Though some potential sperm-specific contraceptive drug targets, such as the calcium channel CatSper, have been identified (*Lishko and Mannowetz, 2018*), our relatively poor understanding of spermatozoon biology makes target-agnostic, or phenotypic strategies more attractive (*Barratt et al., 2017*). Phenotypic drug discovery is undergoing a renaissance in a number of therapeutic areas because it can uncover novel biology in an unbiased way. Retrospective analysis of drug approvals shows the approach to be more successful in first-in-class medicine discovery than previously appreciated (*Swinney and Anthony, 2011*; *Moffat et al., 2017*). However, for male contraceptive drug discovery, a limiting barrier has been the absence of a scalable drug screening system using spermatozoa.

To address this unmet need we set out to develop the first high-throughput phenotypic screening platform for sperm where an imaging-based kinetic analysis module that tracks motility, is followed by a flow cytometry -based assay module that detects both the degree of acrosome reaction (AR) and cell viability - both modules coordinated by a fully automated robotic system. The bespoke computational pipeline quantitatively evaluates each compound's effect on both motility and AR (and viability) thus achieving unprecedented throughput. The platform was used to identify motility inhibitors from a unique 12,000 molecule ReFRAME (Repurposing, Focused Rescue, and Accelerated Medchem) library, which represents the most comprehensive collection of drugs available, as it contains nearly all the small molecules that have achieved regulatory approval and others that have undergone varying degrees of clinical development or have had significant preclinical profiling (*Janes et al., 2018*). This advance opens up the possibility of accelerated male contraceptive development by allowing drug repurposing, target identification as well as screening of chemical diversity libraries for novel medicinal chemistry start points.

## Results

### Phenotypic assay development

Our automated platform (*Figure 1A*) allows for the identification of relatively fast-acting compounds with a rapid plate-processing time (~75 minutes/ 384-well plate), where the motility assay runs for ~30 min followed directly by the AR assay (~45 min). Automated cell tracking (*Figure 1B* and *Figure 1—figure supplement 1A,B*) using time-lapse images allowed kinetic parameter quantification on >200 cells/well (*Figure 1C*). The kinetics show good agreement with clinical systems (*Figure 1—figure supplement 1C*) and a comparable distribution was observed between experimental days, plates and positions (*Figure 1—figure supplement 2*) with any small differences between days stemming from donor pool variance. Importantly, spermatozoa were found to be tolerant to the DMSO concentrations used in our screening program (0.0625%–0.1% *Figure 1—figure supplement 1D*). To determine the optimal screening batch size untreated spermatozoa were dispensed every 30 min for 150 min observing only a small decrease in motility (~10%) after four plates were screened (*Figure 1—figure supplement 1E*). A flow cytometry-based assay measuring AR (*Figure 1D*) was run directly afterwards (*Figure 1E* and *Figure 1—figure supplement 1F*).

### Screening of the ReFRAME library, confirmation of hits by dose response

The ReFRAME library (11,968 compounds) was supplied in 'assay-ready' imaging plates and was solubilised prior to addition of sperm. High assay quality was achieved throughout with a Z'-factor

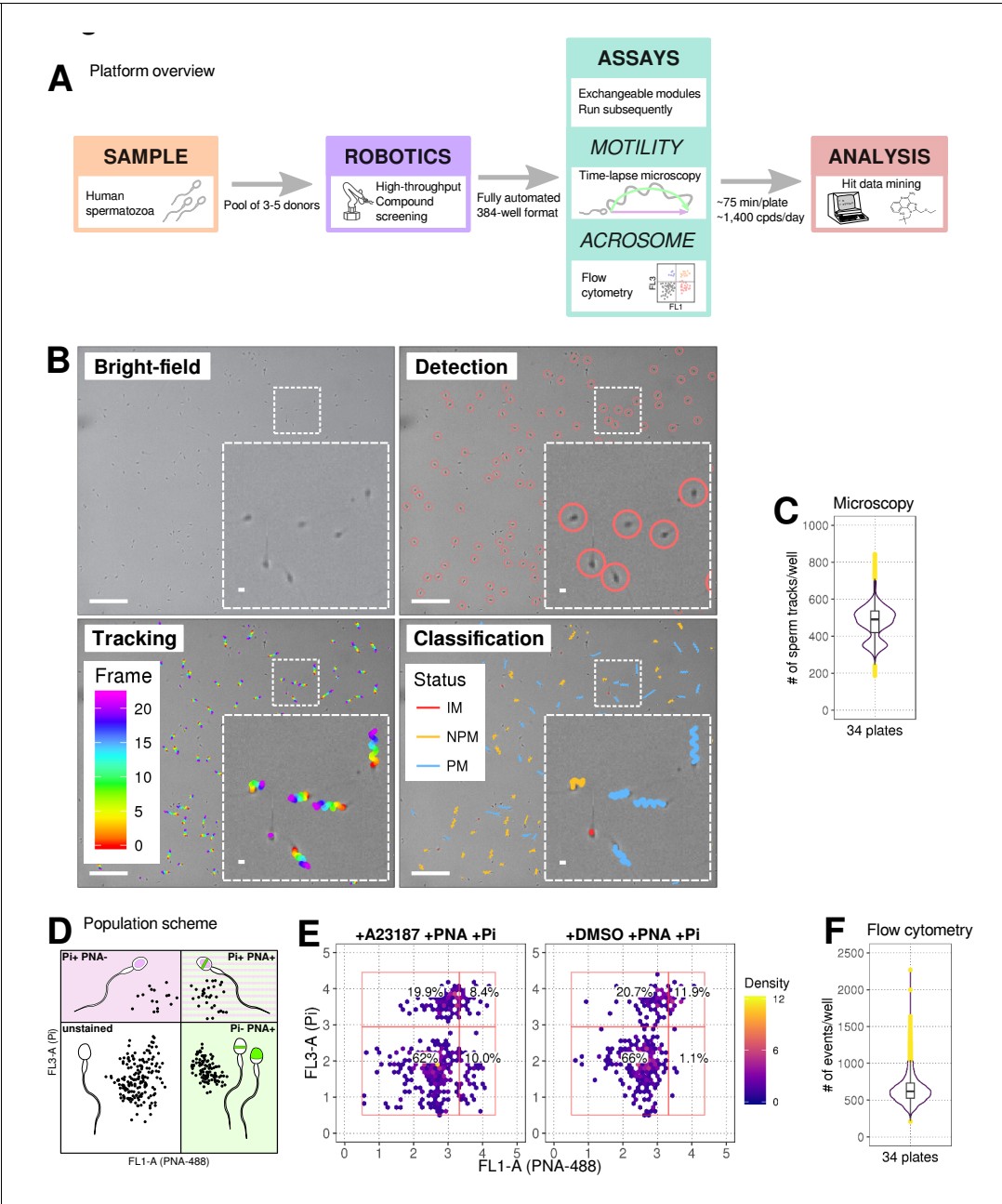

**Figure 1.** Phenotypic Assay workflows. (A) Graphical summary of modular screening workflow where motility measurement is followed by acrosome reaction (AR) measurement allowing a screening throughput of >1400 compounds per donor pool (B) Steps in imaging and analysis: human sperm are recorded with brightfield illumination (first panel) then sperm heads detected using a particle tracking algorithm (second panel) which are then tracked across the timelapse series of images (third panel) and subsequently classified (fourth panel). Each panel contains a zoomed-in subsection of the field. Colour coding for tracking distance and kinetic classification is shown in the panel insets: *Tracking* panel - rainbow gradient (showing progression over time); *Classification* panel - red for immotile (IM), yellow for non-progressively motile (NPM) and blue for progressively motile (PM). Scale bars: 100 µm (main images), 5 µm (insets). (C) Sperm counts per well after microscopy and detection shown in a combined violin/box plot. Colours: purple violin outline (probability density of values), yellow dots (outliers of boxplot). (D) Graphical summary of the expected populations determined by flow cytometry based on distribution of cells measured with FL3-A (Pi) vertical axis and FL1-A (PNA-488) horizontal axis: dead cells (upper left, Pi+ PNA-); dead and acrosome-reacted (upper right, Pi+ PNA+); unstained/live/non-reacted (lower left) and live acrosome-reacted (lower right, Pi-PNA+). (E) Example flow cytometry data comparing sperm treated with the Ca$^{2+}$ ionophore (A23187) which induces AR (left panel) with sperm from DMSO-treated well (right panel). Colours indicate event density. (F) Combined violin/box plot data showing flow cytometry event counts per well. Colours and label as in (C).

The online version of this article includes the following figure supplement(s) for figure 1:

*Figure 1 continued on next page*

*Figure 1 continued*

**Figure supplement 1.** Further characterisation of phenotypic assays.
**Figure supplement 2.** Screening consistency over time analysis.

between 0.4–0.8 (*Figure 2B*). A total of 63 putative hit compounds decreased motility to varying degrees based on a hit selection cut-off of >15% reduction in motility (*Figure 2A* with examples in *Figure 2C,D,F,G* and *Video 1 and 2*). 14 compounds were selected as putative AR+ hits (*Figure 3A* with examples in B, C). Motility hits were confirmed with resupplied material in dose-response experiments. There was a dose-dependent decrease in motility for 29 compounds (46% hit confirmation rate), with $EC_{50}$ values as low as 0.05 µM and effect sizes ranging from 15–100% (*Figure 2F,G*; *Figure 2—source data 1*, *Supplementary file 1*). Amongst the confirmed hits were the aldehyde dehydrogenase inhibitor, Disulfiram (70% maximum reduction at 10 µM) and a putative platelet aggregation inhibitor, KF-4939 (100% max. reduction; $EC_{50}$ = 0.49 µM), and a range of other compounds having very modest effect and/or showing low potency (See *Figure 2—source data 1*, *Supplementary file 1*). In order to test reversibility, wash out experiments were performed. Only a small recovery of motility was observed, after wash out of Disulfiram (*Figure 2—figure supplement 1*). In the AR screen, 9 of the resupplied compounds had a dose-dependent effect with $EC_{50}$ values as low as 0.4 µM (See *Figure 3—source data 2*, *Supplementary file 2*). However, following the orthogonal assay triaging we implemented to eliminate assay interference compounds, none of the AR hits were found to be true positives (see examples in *Figure 3—figure supplement 1D*) despite the primary screen showing good assay robustness.

## Discussion

A major barrier in the search for new male contraceptives has been the lack of an effective high throughput phenotypic screening system. This study presents the first platform based on imaging and flow cytometry that measures two fundamental aspects of sperm behaviour – motility and AR, allowing for the rapid screening of compound collections.

A significant challenge was to create a system for the tracking of these highly motile cells that was fully automated and scalable to allow throughput at sufficient speed. Moreover, concomitant assessment of a second functional attribute, AR along with cell viability was developed in order to maximise use of the biological material. Assessment of motility required novel implementation of tracking algorithms. Importantly, the kinetic outputs were equivalent to those from the 'industry-standard' CASA, which is very low throughput and unsuitable for screening more than a few compounds at one time. Significant workflow optimisation resulted in good assay robustness and an acceptable hit confirmation rate following the primary screen using a medium stringency cut off for hit selection (>15% reduction). For AR detection, a flow cytometry approach was taken based on established methods (*Mortimer et al., 1987*) and showed good assay robustness. However, the presence of assay-interfering compounds in screening libraries (potential fluorescent compounds including DNA intercalators) necessitated further triaging steps (see Materials and methods) to eliminate false positives. This combination of triaging approaches meant that, in this screen, no AR compounds were suitable for progression. An improvement to the AR primary assay could be to use a far-red labelled lectin reagent to detect the glycoproteins. Other variants of the assay would be to either use sperm that had been capacitated, a state of increased readiness to undergo the AR process or use it to find drugs that would block the AR induced by a physiological stimulus. These latter variants are worthy of future investigation.

It is worth noting that whilst the primary goal of this programme is to identify compounds that will act as contraceptive by decreasing motility we detect a larger number of putative motility upregulating hits. These compounds are of interest for not only target identification and uncovering novel sperm biology but also their potential to improve fertility treatments. Overall, although the two assays presented here are fairly complex biological assay using primary cells, it achieved acceptable throughput with the potential to increase batch size for larger screening campaigns.

The advantage of using the ReFRAME collection was the potential to repurpose drugs that are already approved for other indications, or are at earlier stages of clinical progression. The associated

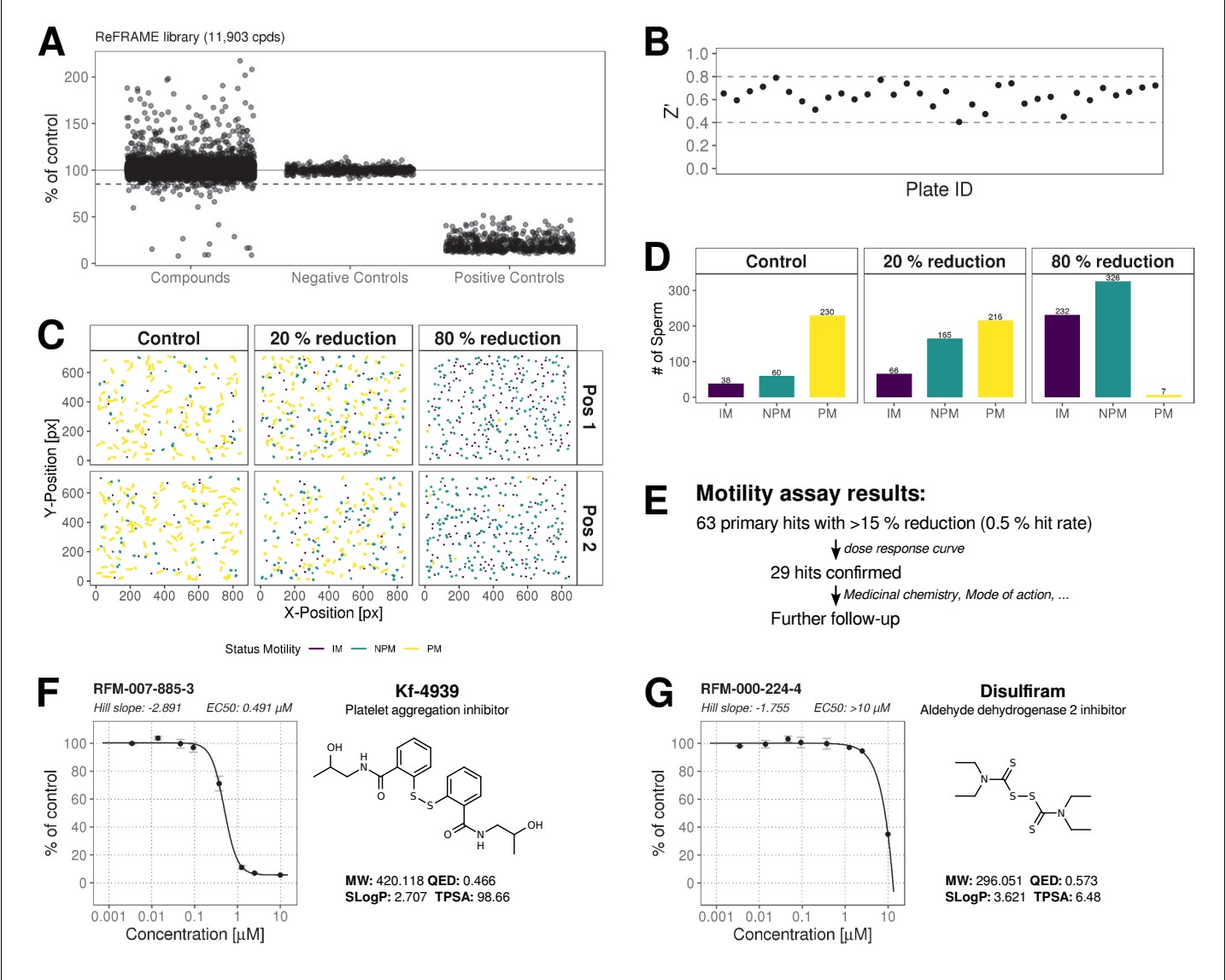

**Figure 2.** ReFRAME Library Screening: Motility Assay Results. (**A**) Primary screening results of the motility assay. Each dot represents a well (either compound or control well) showing % of DMSO control (Curvilinear Velocity: VCL). Positive controls (Pristimerin), negative controls (DMSO) and individual compound datapoints are shown. The dashed line showing the 15% of control (reduction in VCL) – the cut-off for primary hit selection. Total number of compounds = 11,903, excluding wells with auto-focus errors and with 'sticky' compounds which have been excluded from analysis. See dataset (http://doi.org/10.5061/dryad.jdfn2z36z) *Figure 2—source code 1*. (**B**) Assay robustness: the standard high throughput screening metric, Z' (see Materials and methods; *Zhang et al., 1999*) was used to determine the performance of the assay for all screening plates. Dashed lines indicate min/max Z' values. (**C**) Tracking data visualizations of 3 example wells showing sperm tracks of both imaging positions (Position 1 [Pos. 1] and Position 2 [Pos. 2] respectively) within the wells. A DMSO control well (left panels 'Control') shows a large number of progressively motile (PM) sperm (yellow) with few non-progressively motile (NPM) sperm (green and very few immotile (IM) sperm (purple) – this is in contrast to the shorter tracks and higher levels of NPM and IM in the middle panel ('20% reduction') for a compound that shows 20% inhibition of motility (i.e. 80% of control) and the right hand panel ('80% reduction') for a compound showing 80% inhibition (i.e. 20% of control), showing almost all cells are in the IM and NPM classes. (**D**) Histogram of sperm tracks quantification of the data show in (C). (**E**) Summary of motility assay hit rate (0.5%) and reconfirmation rate (0.24%). (**F-G**) Dose response confirmation of two hits. 8-point 3-fold dilution curves are shown with 10 μM as the highest concentration. Two data points per concentration (n = 2, data point is mean ± SD). Each curve is a 4-parameter logistic fit. Each plot shows estimated values Hill Slope and EC$_{50}$. The chemical structure of the hit compound is shown as well as some annotation and physicochemical properties. Physicochemical properties were calculated using RDKit, Python and KNIME: SlogP = partition coefficient (*Wildman and Crippen, 1999*); TPSA is the Topological Polar Surface Area (*Ertl et al., 2000*); MW is the exact Molecular weight; QED = Quantitative Estimate of Drug-likeness (*Bickerton et al., 2012*).

The online version of this article includes the following source data, source code and figure supplement(s) for figure 2:

**Source code 1.** R Code for *Figure 2* primary motility assay.

*Figure 2 continued on next page*

*Figure 2 continued*

**Source code 2.** R Code for *Figure 2—source data 1* dose response confirmation motility assay.
**Source data 1.** Dose response and additional data for primary motility hits.
**Source data 2.** Dose response confirmation data motility assay.
**Figure supplement 1.** Disulfiram Washout CASA measurement of percentage total motility in samples: prior to treatment (Squares); 20 mins after treatment with DMSO (triangles) or 10 µM disulfiram (circles); 60 min after washout of compound/DMSO.
**Figure supplement 1—source code 1.** R code for *Figure 2—figure supplement 1* Disulfiram washout assay.
**Figure supplement 1—source data 1.** Disulfiram washout data.

compound annotations and safety data offers the potential to help accelerate the development of an effective male contraceptive. In the motility assay the hit rate of ~0.2% is typical for a cell-based assay, however surprisingly a relatively high number of hits were not deemed suitable for compound progression. Examples included potentially toxic mercury-containing compounds (phenylmercuric borate and mercufenol chloride); antiseptics and antibiotics including tyrothricin, or compounds with impacts on fundamental biology for example the microtubule stabiliser and chemotherapeutic, docetaxel. However, several hits are potentially of interest. One is Disulfiram (70% reduction at 10 µM) a drug that is well tolerated, has long been used for treating alcohol dependency and has previously been shown to inhibit sperm motility (*Lal et al., 2016*). Disulfiram is unlikely to be developable as a suitable contraceptive drug due to its side-effects when alcohol is consumed but has potential to facilitate mode-of-action studies. One recent report identified 3-Phosphoglycerate dehydrogenase (PHGDH) as a target for disulfiram (*Spillier et al., 2019*) whilst another identified NLP4, a subunit of VCP/p97 segregase (*Skrott et al., 2017*; *Skrott et al., 2019*) although the relevance of either to sperm function is unclear. The identification of KF-4939, a putative platelet aggregation factor (PAF) inhibitor, as a potent hit (100% max. reduction $EC_{50}$ = 0.5 µM) may also be of interest since PAF has been used to improve sperm function in male infertility treatments (*Roudebush et al., 2004*). It is noteworthy that like disulfiram, KF-4939 also contains a disulphide linkage indicating that its biological effect maybe mediated by modifying sulfhydryls on proteins (*Yamada et al., 1985*). Further thiol-containing compounds, bismuth ethanedithiol (98% max reduction; $EC_{50}$ = 2.5 µM) and the anti-fungal benzothiazole, Ticlatone (96% max. reduction $EC_{50}$ = 4.4 µM) were also potent hits, suggesting free cysteines in, as yet, unidentified protein(s), may play a critical role in motility. Of interest is the observation that the Toll-like receptor 7/8 ligand (Resquimod; R848), recently shown to preferentially reduce the motility of X chromosome-bearing mouse sperm by suppressing ATP production (*Umehara et al., 2019*), was a hit, demonstrating that this phenotypic screening approach has the potential to uncover new aspects of sperm biology.

It was somewhat surprising that relatively few drugs or drug-like molecules in the ReFRAME collection were potent and effective inhibitors in the motility assay. This may reflect the unique biology of the spermatozoon that has evolved for a singular purpose – to reach the oocyte and fertilise it and/or that the targets within pharma's therapeutic areas of interest over the last few decades are not similarly functional in human sperm. Conversely however, we did observe a large number of compounds (*Figure 2A*) that had a significant positive effect on sperm motility which may be target agonists as well as potential start points for

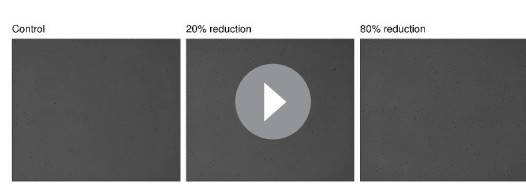

**Video 1.** Movie was generated using a brightfield image sequence for a typical control well (left pane), a well where a compound reduced motility by 20% (middle pane) and one where it reduced it by 80% (right pane).
https://elifesciences.org/articles/51739#video1

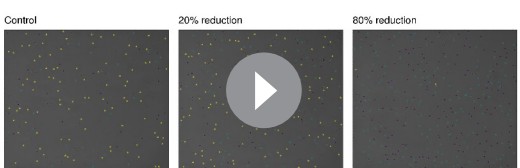

**Video 2.** Movie was generated using a brightfield image sequence for a typical control well (left pane), a well where a compound reduced motility by 20% (middle pane) and one where it reduced it by 80% (right pane). Tracking is overlaid in each panel. Colour coding is detailed in *Figure 2*.
https://elifesciences.org/articles/51739#video2

treating men with low sperm motility. The development of an effective safe male contraceptive is challenging because a prerequisite, other than the obvious need for safety, is a level of reversibility. Other possibilities, such as a drug taken by the female or an agent for use as a topical spermicide (for example to replace nonoxynol-9) are options to consider in the future and depend to a large extent on the pharmacokinetic properties of compounds we identify and their mode of action – for example some may have poor bioavailability.

In summary, our high-throughput phenotypic platform allows for the screening of bioactives including natural products, focussed small molecule libraries and chemical diversity sets, which can provide novel start-points for male contraceptive drug development, as well as the identification of new drug targets in human spermatozoa.

# Materials and methods

## Key resources table

| Reagent type (species) or resource | Designation | Source or reference | Identifiers | Additional information |
|---|---|---|---|---|
| Biological sample (*Homo sapien*) | Live spermatozoa | Donated semen samples | | Local ethical approval (13/ES/0091) |
| Antibody | Lectin PNA (*Arachis hypogaea*), Alexa Fluor 448 Conjugate | ThermoFisher Scientific | ThermoFisher:L21409; RRID: AB_2315178 | Stored as 1 mg/mL stock in DMSO; used at 1:1000 final dilution |
| Commercial assay or kit | Propidium Iodide; Live/Dead Sperm Viability kit, | ThermoFisher Scientific | ThermoFisher:L7011, | Stored as 2.4 mM stock; used at a 1:2000 final dilution |
| Chemical compound, drug | ReFRAME (Repurposing, Focused Rescue, and Accelerated Medchem) library | CALIBR at the Scripps Institute; Publication (*Janes et al., 2018*) | | www.reframedb.org |
| Chemical compound, drug | Pristimerin | Merck | Merck:530070 | Stored as 10 mM stock in DMSO; used at final concentration of 20 µM |
| Chemical compound, drug | calcium ionophore (A23187) | Sigma-Aldrich | Sigma-Aldrich:C7522 | Stored as 10 mM stock in DMSO, used at a final concentration of 10 µM) |
| Chemical compound, drug | Disulfiram | Tocris | Tocris:3807 | Stored as 10 mM stock in DMSO, used at 10 µM final concentration |
| Software, algorithm | Trackpy v0.4.1 | Zenodo. (http://doi.org/10.5281/zenodo.1226458) | | Publication: (*Crocker and Grier, 1996*); Publication: (*Allan, 2018*) |
| Software, algorithm | FFMPEG | FFmpeg Developers (http://ffmpeg.org) | RRID:SCR_016075 | |
| Software, algorithm | Bioconductor packages | Bioconductor (https://bioconductor.org) | RRID:SCR_006442 | flowCore, flowDensity, flowWorkspace, ggcyto |
| Software, algorithm | HDF5 | HDF Group (www.hdfgroup.org) | | |
| Software, algorithm | dr4pl | Dr4pl (https://cran.r-project.org/web/packages/dr4pl/index.html) | | |
| Software, algorithm | Code used for data analysis | This paper | | The R code used for data analysis is included in the supplement files accompanying this paper |

*Continued on next page*

*Continued*

| Reagent type (species) or resource | Designation | Source or reference | Identifiers | Additional information |
|---|---|---|---|---|
| Software, algorithm | KNIME | *Berthold et al., 2008* | | https://www.knime.com |
| Software, algorithm | RDKit | *RDKit, 2018* | | https://www.rdkit.org |

### Ethical approval

Written consent was obtained from each donor in accordance with the Human Fertilization and Embryology Authority (HFEA) Code of Practice (version 8) under local ethical approval (13/ES/0091) from the Tayside Committee of Medical Research Ethics B.

### Development of methods for motility and Acrosome Reaction (AR)

A pre-existing automated cell-based phenotypic screening platform was adapted with the use of live human spermatozoa. The platform utilises a Yokogawa CV7000 Cell Voyager high-throughput microscope able to image 384 multiwell plates under full environmental control acquiring timelapse images at a sufficiently high speed (~48 fps)_to capture the fast-moving sperm for tracking analysis. For high throuput screening with human spermatozoa to be effective and yield meaningful hits, the system needed to replicate, as close as possible, the manual workflow currently used in an

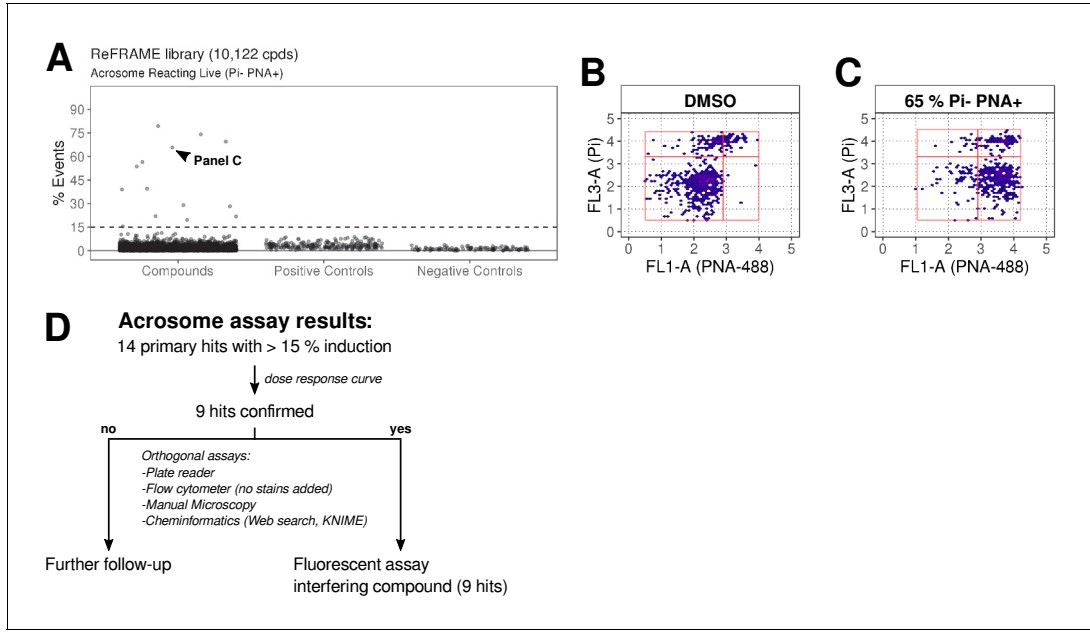

**Figure 3.** ReFRAME library screening: ar assay results. (**A**) Results of primary screening of the library using the acrosome assay (live cells, acrosome reacting, Pi- PNA+ population). Each dot represents a well (either compound or control well). Shown is % Events (number of events in Pi- PNA+ gate relative to total events in the sampled well). Datapoints for compounds, negative controls (DMSO) and positive controls (A23187) are shown. Black dashed line = 15% threshold for primary hits selection. See *Figure 3—source data 1* along with *Figure 3—source code 1*. (**B-C**) Data from two example wells: a DMSO well (left panel) and a well with 65% Pi- PNA+ population (right panel). (**D**) Summary of acrosome assay results before and after triage.

The online version of this article includes the following source data, source code and figure supplement(s) for figure 3:

**Source code 1.** R Code for *Figure 2* primary acrosome assay.
**Source code 2.** R Code for *Figure 3—source data 2* dose response confirmation acrosome assay.
**Source data 1.** Primary screening data acrosome assay.
**Source data 2.** Dose response and additional data for primary AR hits.
**Source data 3.** Dose response confirmation data acrosome assay.
**Figure supplement 1.** Further analysis of AR screening data and triaging strategy.

andrology lab, so that the effect of large numbers of compounds on viable human sperm could be performed. In order to achieve this goal, we investigated a number of factors that could impact the quality of the data such as: plate type; concentration and number of spermatozoa per well; logistics of spermatozoa preparation by density gradient centrifugation and handling; donor-pooling; dispense speed for getting spermatozoa into 384 well plates; timing of compound addition; temperature control; image acquisition parameters; algorithm choice for motility assessment and data management and, evaluating the overall speed of data analysis. In addition, a flow cytometry-based assay and data analysis workflow was developed to measure acrosomal status and cell viability. We chose to perform all experiments on non-capacitated sperm as this condition is the most consistent, less prone to donor-effects and is closer to the state in which the sperm will be when they would be encountering the male contraceptive drug. The goal was to be able to screen a 384-well plate in both assays consecutively within 90 min of sperm dispensing using the fully automatic robotic system. Data was analysed for inter- and intra-plate variation (both technical and biological replicates) and signal-to-background and variance measured to demonstrate the assay was reproducible and robust. The assay was shown to be scalable and all the workflows optimal. Once established, the system was validated by screening the ReFRAME library (*Janes et al., 2018*) and hits confirmed by dose-response analysis.

## Sperm handling

Control semen samples were obtained from volunteer donors with normal sperm concentration, motility and semen characteristics (*WHO, 2010*) and no known fertility problems. Samples were obtained by masturbation after 48–72 hr of sexual abstinence. After ~ 30 min of liquefaction at 37°C, cells were prepared using a discontinuous density gradient procedure (DGC). Semen was loaded on top of a 40–80% suspension of Percoll (Sigma Aldrich, UK) diluted with non-capacitation medium using Minimal Essential Medium Eagle, supplemented with HEPES, Sodium lactate and Sodium Pyruvate to achieve a similar buffer as described previously (*Tardif et al., 2014*). We routinely pooled samples after DGC from 3 to 5 donors for use in each screening batch run in order to reduce donor-to-donor variability. Samples were obtained and analysed in line with suggested guidance for human semen studies where appropriate (*Björndahl et al., 2016*).

## Motility assay

Sperm cells were incubated for 3 hr at 37°C after DGC, transferred to the robotic platform (HighRes Biosolutions Inc) and maintained at 37°C with gentle stirring (100 rpm) in a water bath on top of a magnetic stirrer. Approximately 10,000 spermatozoa (10 μl) were dispensed per well using a Multi-Drop Combi (ThermoFisher) into pre-warmed assay-ready plates. Spermatozoa were incubated with the compounds for 10 mins (thus favouring fast acting compounds) at 37°C in the CV7000 microscope prior to the commencement of imaging.

## Time-lapse imaging

A CV7000 Cell Voyager high-content imaging system (Yokogawa) was used as it allowed full environmental control, sufficient contrast using simple brightfield optics and a fast acquisition rate (up to 45 frames per second). Using a 20x lens with bright field illumination (0.11 ms exposure, 3% lamp power), time-lapse image series were acquired (24 frame in 0.5 s) at two positions per well (with a 400 μm gap between positions to eliminate double counting of sperm). Image acquisition across a 384-well plate with such settings takes ~ 17 min.

## Sperm tracking

A Python implementation of the particle tracking algorithm originally developed by others (*Crocker and Grier, 1996*) was employed (Trackpy v0.4.1 *Allan, 2018*). The algorithm determines the position of every sperm head in each frame and links the position over time creating tracks. We optimized the algorithm parameters for our imaging data to exclude sperm cross-tracks and avoid detecting non-sperm particles. The image data (~17,000 files amounting to 21 GB per plate) can be processed within 30 min using a standard desktop PC (Intel Core i5-6600 CPU, 3.3 GHz, 8 Gb RAM) and can also be parallelized on a compute cluster with minimal effort.

## Acrosome assay

A flow cytometry-based assay was developed employing an iQue Screener (Sartorius) with Alexa488-conjugated peanut agglutinin (PNA) (*Mortimer et al., 1987*) and general sperm viability (using propidium iodide, Pi). This assay was run directly after motility assessment. Controls (DMSO and the $Ca^{2+}$ ionophore A23187) were added using an acoustic dispenser (Echo 555, Labcyte Inc) and incubated for 10 min at 37°C in the SteriStore (HighRes Biosolutions Inc). PNA488 (ThermoFisher Scientific, Cat. No. L21409, stored as 1 mg/mL stock) was then added to achieve a 1:1000 final dilution and propidium iodide (ThermoFisher Scientific, Live/Dead Sperm Viability kit, Cat. No. L7011, stored as 2.4 mM solution) was added to achieve a 1:2000 final dilution using a liquid handler (Tempest, Formulatrix). After addition of dyes, plates were incubated for 10 min at 37°C in the SteriStore before sampling using an iQue Screener (2 s sip time per well with pump setting of 45 rpm). This resulted in a sample processing time of about 25 min per 384-well plate. Cells were categorised as either: unstained cells; 'dead' cells (Pi+ PNA-); Acrosome-reacted but 'dead' cells (Pi+ PNA+); or Acrosome-reacted 'live' cells (Pi- PNA+). This method is capable of detecting a shift of populations (Pi+ and PNA+) upon induction of AR using A23187. The AR assay was performed as an agonist screen (scoring for induction of AR in live cells compared to DMSO controls) with the expectation that compounds could be found that induce AR beyond levels of induction with A23187.

## Controls and QC criteria

For the motility assay DMSO was used as a vehicle control (negative control) and Pristimerin (Merck, Cat. No. 530070, stored as 10 mM stock in DMSO; final concentration of 20 µM) as a positive control. For the acrosome assay we added a calcium ionophore (A23187, Sigma-Aldrich, Cat. No. C7522, stored as 10 mM stock in DMSO, used at a final concentration of 10 µM) as a positive control and DMSO was added to wells as a negative control to be at the same final concentration as that present in the compound-containing wells (0.0625%). DMSO tolerance was tested by performing the standard motility assay exposing sperm to increasing concentrations (max 10%) of DMSO for 10 min. We calculate the standard statistical measure of assay robustness, the Z' value (*Zhang et al., 1999*) for each plate using positive and negative controls for the motility assay and observed Z' values ranging between 0.4–0.8. In addition, we performed a visual check of heatmaps for every plate to detect edge effects.

## Data analysis and normalization

### Motility

custom R scripts were written to calculate standard sperm kinetic parameters (*Mortimer et al., 2015*). Those parameters allow classification of sperm into standard WHO classes: progressively motile (PM) (where average path velocity: VAP > 25 um/sec AND straightness: STR > 80%); non-progressively motile (NPM) (where VAP > 5 um/sec OR straight line velocity: VSL > 11 um/sec), and immotile (IM). In addition to calculating kinetic parameters, we established a workflow to generate movies of time-lapse videos with overlapping sperm tracks using R and FFMPEG (FFmpeg Developers. Available from http://ffmpeg.org). We expressed results as % of control curvilinear velocity: VCL. This was defined as VCL_median (cpd)/VCL_median(DMSO)*100. Hit selection criteria was 15% reduction of VCL. VCL was chosen as the main kinetic measurement as it had an acceptable Z' value and is independent of path averaging. Wells with autofocus errors or compounds which did not dissolve properly (resulting in sperm cells being stuck in one location but moving normally in the rest of the well) were excluded from analysis.

### Acrosome

iQue Screener Data was exported as FCS format and processed using the following *Bioconductor* packages for analysing flow cytometry data: flowCore, flowDensity, flowWorkspace, ggcyto (see https://bioconductor.org). These packages allow handling flow cytometry data as objects, which can then be compensated, gated and visualized efficiently. In addition, we use HDF5 format (www.hdfgroup.org) for storing processed and averaged data. Percent events for each population (Pi+ PNA+, Pi- PNA+, Pi+ PNA- and Pi- PNA-), normalized to total well events, were calculated. Hits were defined as compounds which induce AR beyond 15% (maximum level of induction achieved

with positive control). Wells with irregularities (low in Pi+ or Pi+PNA+ population, or below 200 events) have been excluded from analysis.

## Compound screening

We screened the ~12,000 compound ReFRAME (Repurposing, Focused Rescue, and Accelerated Medchem) library (*Janes et al., 2018*; and www.reframedb.org) supplied by CALIBR at the Scripps Institute. This unique library consists of bioactives and approved drugs that have been assembled from the literature, drug databases and by patent mining. Compounds dissolved in DMSO were spotted (12.5 nL, final assay concentration ~ 6 µM) into 384-well black-sided optical imaging plates (CellCarrier, PerkinElmer) at CALIBR and were shipped on dry ice to the screening centre in Dundee and stored at −20 °C until required. Immediately prior to screening plates were thawed, controls were added using the acoustic dispenser (Echo 555, Labcyte), and 10 µl warm media added followed by shaking on a plate shaker for 10 s. Plates were then incubated for 30 min at 37°C to solubilise the compounds and prewarm the plates prior to the addition of live sperm (see above). With the addition of the media containing sperm (10 µl) the final concentration of DMSO in each well was 0.0625%.

All experiments were performed with pools of donors (3–5 per batch of 4 screening plates) and results normalised to DMSO-treated wells (16 wells per plate). Sperm was first prepared by DGC and then pooled. Where a single donor sample was found to less than 1/3 of the yield required then the contribution from the other doors was increased to compensate.

The ReFrame library has been screened for cytotox (using CellTiterGlo) in HEK293 and HepG2 cells (see www.reframedb.org) with some of the hit compounds displaying activity in one or both assays (*Figure 2—source data 1*), information which helps inform compound progression decisions.

## Dose response experiments

Assay-ready 384-well CellCarrier plates with 8-point curves (10 µM highest concentration) were resupplied by CALIBR in duplicate and processed as described above. Curves were fitted using the R package *dr4pl* (https://cran.r-project.org/web/packages/dr4pl/index.html) using a 4-parameter logistic fit option.

## Triaging of AR hits

Given the prevalence of a range of interfering compounds amongst the library hits, two approaches were used for triaging. These additional triaging steps, after the primary screen, allowed for the elimination of intrinsically fluorescent compounds and compounds fluorescing in the presence of biological material. Screening plates were excited at 488 nm and read at 520 nm and 670 nm in a M1000Pro multimode plate reader (Tecan) after flow cytometry. In addition, when performing dose-response curve follow up on resupply material a replicate set of compound-dosed cells were processed with no-dyes added. This identified the small number of compounds whose fluorescence is indistinguishable from a true positive. In addition, visual confirmation of the appropriate pattern of acrosome staining was performed by microscopy.

## Disulfiram washout assay

In order to assess the ability of sperm cells to recover motility after Disulfiram removal, wash-out experiments were conducted at two timepoints using non-capacitating conditions. Spermatozoa were pooled after DGC preparation in the same manner as for screening. Pooled samples were diluted to a concentration of approximately $10^6$/mL and incubated at 37°C under non-capacitating conditions (see above). After this initial incubation period, Disulfiram (10 µM final concentration) or vehicle control (DMSO to 0.1% final concentration) were added and incubated at 37°C for 20 min. Following exposure to Disulfiram or DMSO control, sperm cells were washed by centrifugation at 300 g for 5 min in 5 mL of the non-capacitation media. After washing, the pellet was resuspended in 500 µl of the appropriate media and incubated at 37°C for 1 hr. Samples were taken for analysis by computer-assisted-sperm analysis (CASA) [CEROS machine (version 12), Hamilton Thorne Research, Beverly, MA, USA] before treatment, after 20 mins treatment and 1 hr post washout. Wash-out experiments were performed on three separated days, utilising different donors to make up the

pooled sample. Each condition was repeated in duplicate on each of these days and each CASA reading was performed in duplicate.

## Acknowledgements

We are very grateful to all members of the research team for their invaluable assistance. We also want to thank all the sperm donors who took part in this study and Dr Stephen Gellatly and Ms Morven Dean for recruitment. We acknowledge the assistance of Irene Sucquart in the early phase of this study. Additionally, we want to thank Dr David Mortimer and Dr Sharon Mortimer for their helpful insights into comparisons with the CASA system and advice in AR assay development. Thanks go to Dr Steve Publicover and Professor C De Jonge for critical reading of the manuscript. Thanks are also due to NPSC lab members for help, particularly John Raynor for engineering support. We thank Mitch Hull, Emily Chen and Kelli Kunen at CALIBR for their help in library plating, logistics and supply of ReFRAME data. Thanks also go to Professor Kevin Reid, Professor Ian Gilbert and Dr Caroline Wilson of Drug Discovery Unit in Dundee for helpful discussions. Special thanks are due to Professor Andrew Hopkins (AH) for his support.

## Additional information

### Competing interests

Christopher LR Barratt: Editor for RBMO, has received lecturing fees from Merck, Pharmasure and Ferring and was on the Scientific Advisory Panel for Ohana BioSciences. The other authors declare that no competing interests exist.

### Funding

| Funder | Grant reference number | Author |
| --- | --- | --- |
| Bill and Melinda Gates Foundation | OPP1160989 | Franz S Gruber<br>Christopher LR Barratt<br>Paul D Andrews |
| Bill and Melinda Gates Foundation | OPP1203050 | Franz S Gruber<br>Zoe C Johnston<br>Christopher LR Barratt<br>Paul D Andrews |

The funders had no role in study design, data collection and interpretation, or the decision to submit the work for publication. The funders facilitated using the REFRAME library

### Author contributions

Franz S Gruber, Resources, Data curation, Software, Supervision, Funding acquisition, Validation, Visualization, Methodology, Project administration; Zoe C Johnston, Data curation, Investigation; Christopher LR Barratt, Conceptualization, Funding acquisition, Methodology; Paul D Andrews, Conceptualization, Resources, Supervision, Funding acquisition, Methodology

### Author ORCIDs

Franz S Gruber https://orcid.org/0000-0003-2008-8460
Zoe C Johnston https://orcid.org/0000-0003-0904-7166
Christopher LR Barratt https://orcid.org/0000-0003-0062-9979
Paul D Andrews https://orcid.org/0000-0001-7699-5604

### Ethics

Human subjects: Written consent was obtained from each donor in accordance with the Human Fertilization and Embryology Authority (HFEA) Code of Practice (version 8) under local ethical approval (13/ES/0091) from the Tayside Committee of Medical Research Ethics B.

Decision letter and Author response
Decision letter https://doi.org/10.7554/eLife.51739.sa1
Author response https://doi.org/10.7554/eLife.51739.sa2

## Additional files

### Supplementary files

• Source data 1. Data of *Supplementary file 1*.

• Source data 2. Data of *Supplementary file 2*.

• Supplementary file 1. Compounds that had a significant effect on sperm motility. Summary of dose response experiments of primary motility hits with estimated EC50 and Efficacy [% reduction] values. Information and names were provided by Calibr. See *Source data 1*.

• Supplementary file 2. Compounds that had a significant effect on Acrosome Reaction. Summary of dose response experiments of primary acrosome hits with estimated EC50 and Efficacy [% increase] values. Information and names were provided by Calibr. Note that none of these compounds passed orthogonal counter screening and are considered as assay interfering compounds/false positives. See *Source data 2*.

• Transparent reporting form

### Data availability

Full data is available. Large files have been deposited to Dryad (http://doi.org/10.5061/dryad.jdfn2z36z).

The following dataset was generated:

| Author(s) | Year | Dataset title | Dataset URL | Database and Identifier |
|---|---|---|---|---|
| Franz S Gruber, Zoe C Johnston, Christopher LR Barratt, Paul D Andrews | 2019 | Data from: A phenotypic screening platform utilising human spermatozoa identifies compounds with contraceptive activity | http://doi.org/10.5061/dryad.jdfn2z36z | Dryad Digital Repository, 10.5061/dryad.jdfn2z36z |

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
