## [Decision Letter]

**Acceptance summary:**

All reviewers agreed that this manuscript reports an important advancement in the development of the tools aimed to provide an unbiased search for novel contraceptives. The authors describe novel drug screening platform to evaluate the libraries of the compounds for their ability to inhibit sperm motility and sperm fertility. The authors demonstrate that such a platform was used successfully to screen for small inhibitory molecules that could potentially serve as templates for further development of novel contraceptive drugs. Thus, this work can have a very high practical impact.

**Decision letter after peer review:**

Thank you for submitting your article "Phenotypic screening platform utilising human sperm for the identification of new compounds with contraceptive activity" for consideration by *eLife*. Your article has been reviewed by three peer reviewers, including Polina V Lishko as the Reviewing Editor and Reviewer #1, and the evaluation has been overseen by Anna Akhmanova as the Senior Editor. The following individual involved in review of your submission has agreed to reveal their identity: Jean-Ju Chung (Reviewer #2).

The reviewers have discussed the reviews with one another and the Reviewing Editor has drafted this decision to help you prepare a revised submission. Our goal is to provide the essential revision requirements as a single set of instructions, so that you have a clear view of the revisions that are necessary for us to publish your work.

Summary:

The reviewers find that the manuscript represents a highly important advancement in the unbiased search for male-based contraceptives. The authors describe the development of a powerful drug screening platform to evaluate compounds for their ability to inhibit sperm motility and sperm fertility. The authors demonstrate that the motility screening platform was used successfully to screen out small inhibitory molecules that could potentially serve as templates for further development of novel contraceptive drugs. However, the small number of hits from the flow-cytometry based AR assays consecutive to the motility screening turn out to be all negative, indicating the system for this aspect remains to be further improved. Overall, the design and details of the current screening platforms are nice additions to the field and the properties of the candidate lead molecules for the motility inhibition will serve a useful foundation for the further endeavor to develop non-hormonal male contraceptives. The data provided are of good quality and the statistical analyses are adequate, while some additional controls will be needed (see below).

Central conclusions:

1) The development of male contraceptives addresses a critical gap in the contraceptive portfolio. Namely, the field is in critical need of a high-throughput system that is robust to variations between men, and to variations within and between plates/trials.

2) The manuscript describes a high-throughput drug-screening platform that can be used to target sperm motility and the sperm acrosome reaction (AR) with smart design of utilizing 12,000 molecules from ReFRAME repurposing library directly applied to human sperm cells from fertile donors.

3) The use of ReFRAME allows both productive uses of compounds that have already undergone lengthy/expensive testing. Of equal importance, it enables compounds to be identified as hits known to affect sperm function or would be predicted to do so.

4) Identification of disulfiram and KF-4939 provide evidence of this important attribute. Other strong positives include the long list of variables they tested to ensure the robustness of the assay.

Overall, the authors provide valuable data validating the approach and showing representative data and are to be commended for revealing the full lists of their hits.

Essential revisions:

The reviewers raise a number of concerns that must be adequately addressed before the paper can be accepted. Some of the required revisions will likely require further experimentation within the framework of the presented studies and techniques.

1) It would be helpful to explain from the beginning the rationale of the assay design and why this assay is focused primarily on the sperm motility and acrosome reaction. To appeal to a scientifically diverse cohort of *eLife* readers, and particularly to people who are not experts in male fertility, the proper explanation and brief introduction to sperm biology is needed.

2) Subsection “Phenotypic Assay Development” and Figure 1—figure supplement 1D: The authors stated that 'spermatozoa were found to be tolerant to DMSO up to 4%.' As the authors note, it is far above typical screening concentration 0.1% DMSO and surprising. Reviewers noted, that to their knowledge, human sperm cells are quite sensitive to DMSO. It would be nice to show the time point of measuring the motility in Figure 1—figure supplement 1D legend. Is the time-course effect of DMSO on sperm motility was tested at given concentrations? It is also recommended to show that the effect of DMSO concentration on the viability (and/or membrane integrity by PI permeability) and its own effect on the AR state of spermatozoa up to 4% DMSO to support the conclusion. Depending on the results, the reviewers recommend that the authors state the sperm tolerance specifically limited to motility or tone down. Additionally, it is needed to see DMSO alone controls on multiple men, and a couple of times, over that full range.

3) The initial screening was performed at ~ 6 μm final concentration and it would be nice to state the concentration of final DMSO concentration per each well to achieve ~ 6 μm and the final concentration of DMSO in the DMSO-treated wells.

4) Figure 2A clearly shows that primary screening results of the motility assay with the ReFRAME library had motility boost hits even if a lot more stringent cut-off is applied. While they were not of interest for a contraceptive development, it could have the potentials for infertility treatments. This reviewer suggests that the authors discuss these 'hits on the opposite side of the spectrum' at a minimum even if the authors decided not to show.

5) Figure 2C and D: do the 20% or 80% reduction in these panels mean the reduction of total motility (PM+NPM) of control? As total sperm counting per each group is different, it will be helpful to indicate the real number of sperm counted on each bar (IM, NPM, and PM) for the readers to easily grasp the calculation.

6) Subsection “Screening of the ReFRAME library, confirmation of hits by dose response” and regarding the negative results of the AR inducer screening: 29 hits were screened out by motility test from the initial 63 positives (~50%) after the dose-response test. In contrast, none of 9 potential candidates from the AR inducer screening were found to be true positives after following the orthogonal assay, which makes the reliability and/or robustness of the AR assays in the screening system in doubt. Can the authors explain the discrepancy in the discovery rate?

7) It would be nice that some alternative strategy is discussed to test the acrosome reaction that won't be hindered by fluorescent compounds. For example, different fluorophores with emissions at more extreme ends of the spectrum could be used, so as not to overlap with the fluorescence spectra of small molecules in the screen used for the flow cytometry. Of course, this will reduce the number of false-positive, but not necessarily the number of the initial hits.

8) Reversibility and less/no side effects are essential requirements for developing or screening male reproductive drugs. Can sperm motility be rescued upon drug elimination?

9) Were there any drugs in the library that are known to have impacts on sperm motility that weren't identified? Failure to detect such a compound might point to a limitation or area in which the assay could be improved.

10) Of the long list of issues that could affect the robustness of the approach, sperm handling is one of the most important. Could more information be provided in the subsection “Sperm handling”, particularly regarding the discontinuous density gradient? Were the samples from different men pooled prior to DGC, or were the washed sperm pooled after? If after, were the samples pooled using equal numbers of sperm from each man, to keep each man's contribution equal, or were all the sperm recovered from each man used? Increased information on this point could help reduce inter-assay variability.

11) Somewhere in the Discussion, authors should provide a paragraph on the potential immediate utility of this approach for identifying topical spermicides/contraceptives. The world does need an improvement over nonoxynol-9, which can increase transmission of STIs, particularly for commercial sex workers. It is worth mentioning that compounds that directly affect sperm function could then be used as a starting point to identify druggable targets that would impair that target during spermatogenesis. Without this discussion, you might be open to criticism that this approach would only work for topicals. There will, of course, be influences of metabolism, etc., but high throughput screening for targets is a novel and critically important contribution.

12) Acrosome reaction was used as one of the key parameters for screening, and yet the assessed samples of sperm cells were ejaculated spermatozoa and not the capacitated cells. According to common knowledge, sperm ability to undergo an acrosome reaction is achieved only after they are capacitated (in vivo or in vitro), and yet in this study, the capacitation was not performed (subsection “Sperm handling”). To clarify this confusion and make the text more digestible to readers, authors are encouraged to expand their very brief result and/or Discussion sections. Specifically, since the goal of the team was to search for male contraceptives, it is not clear whether the compounds are expected to target sperm while they are stored inside the male reproductive tract or post-ejaculated. As reviewers noted, the compounds that prematurely induce acrosome exocytosis could only serve as topical contraceptives. Whether in a condom, jelly, or foam, that would come into contact with only non-capacitated sperm. Premature induction of exocytosis would NOT be recommended as a means of killing the sperm inside male reproductive tract, because if they were in the rete or epididymis, that would result in a sperm granuloma and obstructive azoospermia. This might be a point worth mentioning in the Discussion.

---

## [Author Response]

Essential revisions:The reviewers raise a number of concerns that must be adequately addressed before the paper can be accepted. Some of the required revisions will likely require further experimentation within the framework of the presented studies and techniques.1) It would be helpful to explain from the beginning the rationale of the assay design and why this assay is focused primarily on the sperm motility and acrosome reaction. To appeal to a scientifically diverse cohort of eLife readers, and particularly to people who are not experts in male fertility, the proper explanation and brief introduction to sperm biology is needed.

This is a very good suggestion – we acknowledge that in the desire to keep the manuscript short we neglected to set the scene and explain the logic behind our approach. We have thus expanded the second paragraph of the Introduction.

2) Subsection “Phenotypic Assay Development” and Figure 1—figure supplement 1D: The authors stated that 'spermatozoa were found to be tolerant to DMSO up to 4%.' As the authors note, it is far above typical screening concentration 0.1% DMSO and surprising. Reviewers noted, that to their knowledge, human sperm cells are quite sensitive to DMSO. It would be nice to show the time point of measuring the motility in Figure 1—figure supplement 1D legend. Is the time-course effect of DMSO on sperm motility was tested at given concentrations? It is also recommended to show that the effect of DMSO concentration on the viability (and/or membrane integrity by PI permeability) and its own effect on the AR state of spermatozoa up to 4% DMSO to support the conclusion. Depending on the results, the reviewers recommend that the authors state the sperm tolerance specifically limited to motility or tone down. Additionally, it is needed to see DMSO alone controls on multiple men, and a couple of times, over that full range.

We appreciate the reviewers concern about DMSO sensitivity on spermatozoa. We have reworded and added to the text e.g. subsections “Phenotypic Assay Development”, “Controls and QC criteria”, “Compound Screening”

and updated Figure 1—figure supplement 1D, to highlight the DMSO concentrations used in the screen and hit confirmation. We have also added data in Figure 1—figure supplement 1D for DMSO concentrations up to 10 μM, where we start to see an effect on sperm motility in our assay (10-30 min incubation with compound/DMSO). We did not want the reader to get the impression that spermatozoa are tolerant to high concentrations of DMSO, but wanted to point out that with the range of concentrations used in our assay (up to 0.1% DMSO) we did not see an effect caused by DMSO over the time course of screening. Further evidence to support this is provided by the newly added wash out experiments performed on three different donor pools using 20 min incubation of either 0.1% DMSO or 10 μM Disulfiram (new Figure 2—figure supplement 2).

3) The initial screening was performed at ~ 6 μm final concentration and it would be nice to state the concentration of final DMSO concentration per each well to achieve ~ 6 μm and the final concentration of DMSO in the DMSO-treated wells.

The final concentration of DMSO used in the primary screen was 0.0625% and was the same in both compound wells and control wells. This is detailed in the Materials and methods subsection “Controls and QC criteria” and also mentioned in the subsection “Compound Screening”.

4) Figure 2A clearly shows that primary screening results of the motility assay with the ReFRAME library had motility boost hits even if a lot more stringent cut-off is applied. While they were not of interest for a contraceptive development, it could have the potentials for infertility treatments. This reviewer suggests that the authors discuss these 'hits on the opposite side of the spectrum' at a minimum even if the authors decided not to show.

We agree it would be useful for the readers to appreciate the dual purpose. The characterisation of these hits was outside the scope of the contraceptive programme but clearly are of interest. We have added two sentences highlighting this in the Discussion (third and fifth paragraphs).

5) Figure 2C and D: do the 20% or 80% reduction in these panels mean the reduction of total motility (PM+NPM) of control? As total sperm counting per each group is different, it will be helpful to indicate the real number of sperm counted on each bar (IM, NPM, and PM) for the readers to easily grasp the calculation.

We have adjusted Figure 2D to include the numbers. The percentage values shown were indeed total motility expressed as% of control. For this we use a median VCL value of all spermatozoa tracked over two positions in each well. This median is normalized to a median VCL value of DMSO control wells. Classifying sperm tracks by WHO definitions, we can then count each group and see what changes occur upon compound addition e.g. 20% reduction shows an increase in immotile and non-progressively motile classified cells, while progressively motile-classified cells are reduced in number.

6) Subsection “Screening of the ReFRAME library, confirmation of hits by dose response” and regarding the negative results of the AR inducer screening: 29 hits were screened out by motility test from the initial 63 positives (~50%) after the dose-response test. In contrast, none of 9 potential candidates from the AR inducer screening were found to be true positives after following the orthogonal assay, which makes the reliability and/or robustness of the AR assays in the screening system in doubt. Can the authors explain the discrepancy in the discovery rate?

For the motility data we took a fairly low cut off (>15% decrease in motility) in the primary screen to ensure we identified compounds with a range of effect sizes and thus more start points for medicinal chemists to evaluate. Inevitably, some of those weakly downregulating compounds (~50%) did not confirm in dose-response experiments.

For the AR assay we chose a hit cut off based on the median effect size of our positive control A23187 (15%). Putative hits were then selected and 9/14 reconfirmed (64%). On this basis and examining plate metrics the assay is robust. However, the primary AR assay has two limitations: Firstly, the presence of fluorescent molecules gives rise to false positives in the primary screen. However, given the low number of primary hits and the relative accessibility of the assay, it is possible to screen out false positives in orthogonal assays with little effort. As shown in Figure 3—figure supplement 2D some of these “hit” molecules were compounds that fluoresce in the presence of biological material. The platform has the potential to discover compounds increasing acrosome reaction, but this was our first screen with a medium sized drug collection of limited availability. Variations on the assay could potentially offer improvements – for example changing fluorophore (as the reviewers point out in point 7 below), or offer alternative outcomes – such as screening spermatozoa that had been incubated under capacitating conditions, or finding compounds that block induction of the AR.

7) It would be nice that some alternative strategy is discussed to test the acrosome reaction that won't be hindered by fluorescent compounds. For example, different fluorophores with emissions at more extreme ends of the spectrum could be used, so as not to overlap with the fluorescence spectra of small molecules in the screen used for the flow cytometry. Of course, this will reduce the number of false-positive, but not necessarily the number of the initial hits.

We have considered this post hoc and indeed this approach should decrease the primary hit numbers but as mentioned, not necessarily the number of true positives. We have added a sentence suggesting a possible modification (Discussion, second paragraph). Other alternatives would include a microscope-based assay using fixed cells, however this is hard to combine with a live cell assay.

8) Reversibility and less/no side effects are essential requirements for developing or screening male reproductive drugs. Can sperm motility be rescued upon drug elimination?

We agree with the reviewers that the safety profile of a contraceptive is critical. As is the case for all drug development projects the toxicological side effects will need to be scrutinized further downstream and would be dependent on how the compound was administered and to whom. Indeed, reversibility is an important issue to consider and understand. Irreversible inhibitors are often highly effective drugs. In the case of a contraceptive, such a mode of inhibition would be predicted to resist natural “wash out” as the spermatozoon moves from the male to the female reproductive tract. The downside would be that to restore fertility, time would need to pass – possibly for as long as the full spermatogenic cycle. In response to the reviewers’ comments, we have investigated reversibility using Disulfiram and have shown the data from a series of wash-out experiments in Figure 2—figure supplement 2. This data indicates that the negative effect of Disulfiram on sperm motility remains after washing out (measured at 60 min after a recovery phase). We have modified the Results (subsection “Screening of the ReFRAME library, confirmation of hits by dose response”) as well as Materials and methods section (“Disulfiram washout assay”) accordingly. We have added a fourth author to reflect their input into the additional experiments that were performed – all authors agree with this addition.

9) Were there any drugs in the library that are known to have impacts on sperm motility that weren't identified? Failure to detect such a compound might point to a limitation or area in which the assay could be improved.

This is an important point. Unfortunately, as part of the MTA with Calibr we only get to see the structures of the confirmed hits (up to a maximum of 1% of the library) so we cannot directly answer this question. False negatives are a concern in high-throughput screening. The presence of false negatives can be due to limitations in the assay or stability/quality of the compounds, or by choosing a high hit selection cut-off. Other libraries that can be screened in the future will hopefully address this point.

10) Of the long list of issues that could affect the robustness of the approach, sperm handling is one of the most important. Could more information be provided in the subsection “Sperm handling”, particularly regarding the discontinuous density gradient? Were the samples from different men pooled prior to DGC, or were the washed sperm pooled after? If after, were the samples pooled using equal numbers of sperm from each man, to keep each man's contribution equal, or were all the sperm recovered from each man used? Increased information on this point could help reduce inter-assay variability.

We agree this should have been clarified. The samples were pooled after DGC and in cases were one donor was of lower yield (<1/3 of what was needed) the contribution from the other donors was increased. Pool size was between 3-5 donors/day. We have added text added in the Materials and methods (subsection “Compound screening”) to help clarify this.

11) Somewhere in the Discussion, authors should provide a paragraph on the potential immediate utility of this approach for identifying topical spermicides/contraceptives. The world does need an improvement over nonoxynol-9, which can increase transmission of STIs, particularly for commercial sex workers. It is worth mentioning that compounds that directly affect sperm function could then be used as a starting point to identify druggable targets that would impair that target during spermatogenesis. Without this discussion, you might be open to criticism that this approach would only work for topicals. There will, of course, be influences of metabolism, etc., but high throughput screening for targets is a novel and critically important contribution.

We agree that this is worthy of further discussion. We have added a paragraph (Discussion, fifth paragraph). As suggested, not every active agent will be suitable for a drug that can be taken by the male.

12) Acrosome reaction was used as one of the key parameters for screening, and yet the assessed samples of sperm cells were ejaculated spermatozoa and not the capacitated cells. According to common knowledge, sperm ability to undergo an acrosome reaction is achieved only after they are capacitated (in vivo or in vitro), and yet in this study, the capacitation was not performed (subsection “Sperm handling”). To clarify this confusion and make the text more digestible to readers, authors are encouraged to expand their very brief result and/or Discussion sections. Specifically, since the goal of the team was to search for male contraceptives, it is not clear whether the compounds are expected to target sperm while they are stored inside the male reproductive tract or post-ejaculated. As reviewers noted, the compounds that prematurely induce acrosome exocytosis could only serve as topical contraceptives. Whether in a condom, jelly, or foam, that would come into contact with only non-capacitated sperm. Premature induction of exocytosis would NOT be recommended as a means of killing the sperm inside male reproductive tract, because if they were in the rete or epididymis, that would result in a sperm granuloma and obstructive azoospermia. This might be a point worth mentioning in the Discussion.

As the reviewers point out acrosome reaction is normally investigated using capacitated sperm. Sperm capacitation in the laboratory is achieved after 2-3 hours under specific buffer conditions. The work sponsored by the BMGF was to examine the use of a compound administered to the man to affect sperm function. As such we used non-capacitating conditions, as the cells in the epididymis – where the potential site of action is proposed – would not be in a capacitated state. In our platform, non-capacitated sperm cells are incubated with compounds, motility is measured, and subsequently acrosome status is measured. This offers the potential to find drugs or compounds that induce acrosome reaction in non-capacitated cells. In a different screening workflow, one could aim to find molecules which block acrosome reaction (provided we could obtain natural agonists of the AR e.g. ZP). The AR assay could also readily be run “stand-alone” on capacitated cells.

Premature acrosome reaction in the male reproductive tract could indeed cause problems and this would of course need to be investigated. However, it is interesting to note that when men have a vasectomy and all the sperm cells are resorbed into the body (spermatogenesis continues) fertility returns in the overwhelming majority of men upon reversal indicating that even in extreme circumstances destruction of the spermatozoa is not a significant issue.